# Development of the parental self-efficacy scale for preventing challenging behaviors in children with autism spectrum disorder

**Yuri Kabashima**[id]*, **Etsuko Tadaka, Azusa Arimoto**[id]

Department of Community Health Nursing, Graduate School of Medicine, Yokohama City University, Yokohama, Kanagawa, Japan

* honu.sun.sora@gmail.com

## Abstract

**Data Availability Statement:** All relevant data are within the manuscript and its Supporting Information files.

**Funding:** The author(s) received no specific funding for this work.

### Background

Almost all children with autism spectrum disorder (ASD) have experienced challenging behavior, including disruptive and aggressive behavior symptoms to both themselves and others. In conjunction with appropriate strategic parenting, challenging behavior can be prevented by empowering children's sociality and optimizing their environment. However, a means of measuring such parenting has yet to appear. This study developed the Parental Self-Efficacy Scale for Preventing Challenging Behaviors in Children with Autism Spectrum Disorder (PASEC) and evaluated its reliability and validity.

### Method

Self-administered questionnaires were distributed to 1,344 parents of children with ASD at all 521 child development support centers in Japan. Confirmed construct validity of the PASEC was determined using confirmatory factor analysis. Internal consistency of the PASEC was calculated using Cronbach's alpha. The self-efficacy subscale of the Parenting Sense of Competence (PSOC) was administered to assess criterion-related validity of the PASEC.

### Results

In total, 260 parents provided valid responses. Exploratory and confirmatory factor analyses identified six items from two factors: empowerment of children's sociality and optimization of children's environment. The final model showed goodness-of-fit index, 0.981; adjusted goodness-of-fit index, 0.944; comparative fit index, 0.999; and root mean square error of approximation, 0.019. Cronbach's alpha for the entire PASEC was 0.82; that for each factor was above 0.70. The correlation coefficient between the self-efficacy subscale of the PSOC and the entire PASEC was $r = 0.52$ ($P < 0.001$).

**Competing interests:** The authors have declared that no competing interests exist.

## Conclusions

The PASEC demonstrated adequate reliability and validity to assess parents' self-efficacy for preventing challenging behavior for children with ASD. That scale can help prevent challenging behavior; it can contribute to improving the mental health of parents and children with ASD as well as to primary prevention of child maltreatment and abuse.

## Introduction

Autism spectrum disorder (ASD) refers to persistent deficits in social communication and social interaction as well as restricted repetitive behaviors, interests, and activities with or without intellectual disability [1]. Matson et al. [2] found that 94% of parents having children with ASD reported that the children had "challenging behavior". Challenging behavior is defined as "culturally abnormal behaviour of such an intensity, frequency or duration that the physical safety of the person or others is likely to be placed in serious jeopardy, or behaviour which is likely to seriously limit use of, or result in the person being denied access to, ordinary community facilities" [3]. Challenging behavior includes disruptive and aggressive behavior symptoms to both oneself and others [4, 5]. One study showed that parents' perceptions of difficult behavior in their children are associated with increased risk of parental stress and abuse [6]. Parents of children with ASD are more stressed than those with typically developing children and children with other developmental disabilities [7–9]. Children with ASD have a higher risk of maltreatment than those with typical development [10]. Therefore, toward improving the mental health of parents and children with ASD as well as toward primary prevention of maltreatment and abuse, it is important to prevent the challenging behavior of children with ASD.

The challenging behavior of children with ASD is related to their sociality and influenced by their environment; it can be prevented by empowering children's sociality and optimizing their environment. Factors for challenging behavior include the following: lack of stimulation; restricted access to preferred items and activities; low attention; social constraints and aversive stimuli; and developmental characteristics, such as ASD [11]. Children with ASD have disabilities in social communication and social interaction [1, 12]; almost half of them have language disabilities [13]. Social interaction disabilities in children with ASD have been found to be associated with challenging behavior [2]. Emerson [3] stated that challenging behavior is likely to occur in certain environment and result from human-environment interactions. Children with ASD have unique sensibilities, such as hypersensitivity, and feel some stimulation that other children do not [14]. Thus, to express their feelings when they feel some kind of stimulation from the environment, children with ASD respond by challenging behavior, rather than verbal expressions. Therefore, appropriate strategic parenting, such as empowering children's sociality and optimizing their environment, can prevent challenging behavior.

To prevent challenging behavior, self-efficacy is a key concept toward promoting the behaviors of parents having children with ASD. Bandura [15] defined self-efficacy as one's belief in one's ability to succeed in specific situations or accomplish a task and stated that it affects the behavior required to produce results. Most parents do not know how to respond to children's challenging behavior [16]. Parents feel stressed by challenging behavior but are unable to receive any support [17]. Hence, if parents can enhance their self-efficacy in promoting prevention of their children's challenging behavior, they can promote their empowerment in response to such behavior. Thereby, parents can respond to their children's feelings, gain an

opportunity to secure peace of mind, and ensure their children's physical safety. Consequently, parents will acquire, maintain, and improve opportunities for their children's social involvement and promoting their growth and development. Further, improving parents' self-efficacy to prevent challenging behavior will improve the mental health of both parents and children as well as prevent maltreatment and abuse.

The concept of self-efficacy for promoting prevention of challenging behaviors has not, however, yet been clarified and assessed. Measures of parenting self-efficacy developed in previous studies have limitations. First, the primary targets of existing parenting self-efficacy scales are not parents of children with ASD but those of typically developing children [18]. Second, existing parenting self-efficacy scales are not task-specific, such as prevention of challenging behavior of parents, but domain-specific, such as parenting of parents [19], in the three types of self-efficacy: general, domain-, and task-specific [18]. Through parental self-efficacy scales to prevent challenging behaviors, practitioners can achieve the following: measure parents' self-efficacy scores for preventing challenging behaviors; identify parents with low self-efficacy scores; and support such parents in improving their self-efficacy to prevent challenging behaviors. Increased parental self-efficacy enables parents to take action to prevent challenging behaviors: that can prevent children's challenging behaviors, improve the mental health of parents and children with ASD, and primarily prevent child maltreatment and abuse. One parental self-efficacy (PSE) scale, the Early Intervention Parenting Self-Efficacy Scale [20], consists of two factors: Parent Outcome Expectations; and Parent Competence. Parent Outcome Expectations relates to caregivers perceiving themselves as effective in parenting their children. Parent Competence relates to the belief that the outcome of ASD is a function of environmental influences or constraints, such as family background. However, parents of children with ASD have subject-specific challenges that make it difficult to adapt existing versions of the PSE scale owing to differences in concepts. The subject-specific challenges for parents of children with ASD are that the children have communication difficulties; the children are also hypersensitive or insensitive to stimuli from the surrounding environment, which induce challenging behaviors.

Thus, children with ASD need more communication experiences to form patterns of communication and develop social skills; to reduce their children's discomfort, the parents need to adjust the stimuli that are uncomfortable for their children as well as positive stimuli. To that end, we believe that the Parental Self-Efficacy Scale for Preventing Challenging Behaviors in Children with Autism Spectrum Disorder (PASEC) can measure parents' self-efficacy scores for preventing challenging behaviors. Therefore, a new scale, PASEC, needs to be developed.

In the present study, we developed a scale for parents of children with ASD to assess the parent's self-efficacy for preventing challenging behaviors. We developed the PASEC and examined its reliability and validity. Parental self-efficacy for preventing challenging behaviors in children with ASD is defined as the belief that parents can empower children's sociality and optimize their environment toward preventing their children's challenging behavior.

## Materials and methods

### Phase 1: Developing the instrument

First, we developed a pool of items based on a literature review. From the perspective of the parents' behavior regarding to prevent challenging behavior, we searched PubMed, Web of Science, and Ichushi-web for related articles. We did so use specific keywords: challenging behavior; problem behavior; autism spectrum disorder; preschool; parents; environment; and coping. In this way, we identified 10 articles [14, 21–29]. We based the item inclusion criteria on three perspectives: (1) prevention of challenging behavior by the parents; (2) parents'

behavior; and (3) practically beneficial items. Using those viewpoints, we reviewed the pool of draft items and made several modifications; the result was a final list of 23 items.

Second, the pool of items was reviewed by three professionals, two researchers, and three mothers. The professionals comprised two public health nurses and a facility manager for child development support. The researchers specialized in community health nursing and had studied parents of children with developmental disabilities. The mothers had preschool children. They assessed the content validity, face validity, and practical usefulness of the items. Following the reviewers' opinions, we revised the wording of each item. As a result, the modified scale was reduced to 13 items. Each item was scored on a four-point Likert scale, ranging from 0 (disagree) to 3 (agree).

## Phase 2: Validating the instrument

**Study participants.** This survey covered 1,344 parents of children with ASD at all 521 child development support centers in Japan. We obtained details of these centers from publicly available information lists. Before sending the survey questionnaires to each center, we calculated the necessary number of questionnaires from the number of preschool children and centers within each prefecture.

Informed consent for this study was obtained in two steps: at the centers and with the parents. First, we sent a letter to the centers explaining the purpose of the study. We also asked the managers of the centers to assess the ability to consent and to select suitable parents as potential subjects in line with the study's inclusion criteria. Second, the center managers gave the parents a verbal explanation and handed them request letters, detailing the purpose of the study and stating that participation was entirely free. The managers also handed out questionnaires. In this study, assessment of ASD characteristics was not conducted by a doctor or psychologist but by the center managers or the children's parents. Of the participants contacted, 264 (19.6%) responded; of those, 260 (98.5%) provided questionnaires with valid responses suitable for analyses.

**Measures.** The parents' demographic characteristics included age, sex, household status, employment status, type of employment, education level, perceived health, and prevalence of depression (Table 1). We used a visual analog scale to assess overall perceived health [30]: respondents were asked to write a dot on a 0–100 scale to reflect their overall perceived health. The score of 0 represents the worst possible health a person can have; 100 signifies perfect health. This scale correlates with the score of a Japanese version of subjective well-being.

We used a Japanese version of the K6 to assess the mental health of the parents [31]. The K6 comprises six items, which rate the frequency of distress symptoms from 0 (never) to 4 (always). We used the total score of six items (range, 0–24) for the respondent characteristics. The depression cutoff was 5 or higher; thus, if the total score was higher than 5, it was rated as depression [32]. This scale demonstrated excellent areas under the curves.

The demographics of children with ASD included age, sex, birth order, number of children in family, and degree of challenging behavior (Table 2). We assessed the degree of challenging behavior using a Japanese version of the Aberrant Behavior Checklist (ABC) [33, 34]. The ABC comprises 58 items divided into five subscales: Irritability; Lethargy; Stereotypy; Hyperactivity; and Inappropriate Speech. Of those subscales, we used only Irritability (ABC-I) because we judged that the subscale represented a characteristic state of challenging behavior noticed by parents. The item of Irritability (ABC-I) includes destructive and aggressive behavior toward oneself or others as the challenging behavior [4, 5]. Destructive and aggressive behavior is commonly recognized by the parents of children with autism as a challenging behavior [17]. The ABC-I has also been used as an indicator of behavioral problems, which are

**Table 1. Demographic characteristics of parents.** n = 260.

| | | n or Mean±SD or Median | % or range or Interquartile range |
|---|---|---|---|
| Age (years) | | 38.2±5.0 | 26.0–59.0 |
| | Missing | 2 | 0.8 |
| Sex | Female | 241 | 92.7 |
| | Male | 19 | 7.3 |
| Household status | Parents | 217 | 83.5 |
| | Parents and grand parents | 33 | 12.7 |
| | Single parent | 5 | 1.9 |
| | Other | 5 | 1.9 |
| Employment status | No | 139 | 53.5 |
| | Yes | 121 | 46.5 |
| Type of employment | Part time employment | 68 | 56.2 |
| | Full time employment | 39 | 32.2 |
| | Other employment | 13 | 10.8 |
| | Missing | 1 | 0.8 |
| Education level | More than college/university | 98 | 37.7 |
| | Vacational school /Junior college | 84 | 32.3 |
| | High school | 73 | 28.1 |
| | Junior high school | 4 | 1.5 |
| | Missing | 1 | 0.4 |
| Perceived health (scores) | | 70.0 | 50.0–80.0 |
| | Missing | 14 | 5.4 |
| Prevalence of depression: K6 (scores ≧5) | | 114 | 43.8 |
| | Missing | 7 | 2.7 |
| Parenting self-efficacy: PSOC (scores) | | 15.3±5.3 | 6.0–32.0 |
| | Missing | 1 | 0.4 |

PSOC: Self-efficacy subscale of the Parenting Sense of Competence scale

SD: Standard deviation

similar to challenging behavior [35]. That subscale consists of 15 items (e.g., injures self, aggressive to others, screams inappropriately); the responses are on a four-point Likert-type scale, ranging from 0 (no problems) to 3 (severe problems). The total score of the subscale ranges from 0 to 45, with higher scores indicating more challenging child behavior. The subscale had a Cronbach's alpha of 0.92 [34].

To assess the construct validity of the PASEC, participants completed the Japanese version of the Parenting Sense of Competence (PSOC) subscale [36, 37]. The PSOC comprises two subjective scales that measure parenting satisfaction and parenting self-efficacy. Of these subscales, we used only that for parenting self-efficacy: the content was relevant to the PASEC in terms of including parents of children with developmental disability and self-efficacy related to parenting. That subscale consists of six items (e.g., being a parent is manageable, and any problems are easily solved); responses are on a six-point Likert-type scale, ranging from 1 (strongly disagree) to 6 (strongly agree). The total score of the subscale ranges from 6 to 36, with higher scores indicating greater parenting self-efficacy among parents. This PSOC scale had a Cronbach's alpha of 0.79; the self-efficacy subscale had a Cronbach's alpha of 0.76 [37].

**Ethical considerations.** This study was conducted with the approval of the Institutional Review Board of the Medical Department of Yokohama City University School (Approval No. A190700009).

**Table 2. Demographic characteristics of children with autism spectrum disorder.** n = 260.

| | | n or Mean±SD | % or range |
|---|---|---|---|
| Age (years) | | 4.7±1.1 | 2.0–6.0 |
| Sex | Male | 196 | 75.4 |
| | Female | 63 | 24.2 |
| | Missing | 1 | 0.4 |
| Birth order | First | 131 | 50.3 |
| | Second | 81 | 31.2 |
| | Third | 39 | 15.0 |
| | Fourth | 7 | 2.7 |
| | Missing | 2 | 0.8 |
| Number of children | One | 78 | 30.0 |
| | Two | 110 | 42.3 |
| | Three or more | 72 | 27.7 |
| Challenging behavior: ABC-I (scores) | | 10.3±8.7 | 0.0–43.0 |
| | Missing | 4 | 1.5 |

ABC-I: Irritability subscale of the Aberrant Behaviour Checklist

SD: Standard deviation

**Statistical analyses.** We conducted all analyses using IBM SPSS Statistics 25.0 and Amos 24.0 (Chicago, Illinois, USA). We undertook item analysis to investigate the reliability of the scale and exploratory factor analysis to investigate the factor structure of the scale. The criteria for item analysis included distribution ("agree" and "agree a little" were over 85%; kurtosis and skewness were over ± 1.0), rates of response difficulty (non-respondents ≥5%), correlations between each item (correlation coefficient >0.6), item-total analysis (correlation coefficient <0.3), and good-poor analysis (no significant differences between the highest- and lowest-scoring groups).

We randomly divided the total sample (n = 260) into two sub-samples for cross-validation: group 1 (n = 130) for performing exploratory factor analysis; and group 2 (n = 130) for performing confirmatory factor analysis. We examined the items remaining after item analysis using exploratory factor analysis (principal factor method) with promax rotation. With reference to eigenvalues and scree plots, we estimated that there were one to two factors. We then repeated the exploratory factor analysis, assuming one to two factors and excluding items with item loadings <0.5. We determined factor reliability according to a Cronbach's alpha ≥0.7; construct validity was verified with confirmatory factor analysis. We examined model fit using the goodness-of-fit index (GFI), adjusted GFI (AGFI), comparative fit index (CFI), and root mean square error of approximation (RMSEA). The model was accepted if the GFI and AGFI were ≥0.90, CFI was ≥0.95, and RMSEA was ≤0.05 [38, 39]. We also examined construct validity by the correlation between total score of the PASEC and total score of self-efficacy subscale of the PSOC. We evaluated a correlation of ≥0.50 as adequate [40]. In addition, we assumed that the total PASEC score and total ABC-I score (indicating the degree of challenging behavior of children) and total K6 score (indicating the mental health of parents) would have a negative correlation. Therefore, we calculated the correlation coefficients to clarify the relationships.

## Results

### Respondent characteristics

Table 1 shows the demographic characteristics of the parents. The parents' mean age was 38.2 years. In all, 92.7% were female; 83.5% were living with their spouses and children; 46.5% were employed.

Table 2 shows the demographic characteristics of children with ASD. In all, 29.6% of the children were aged 6 years; 75.4% were boys. The children's mean score of challenging behavior (ABC-I) was 10.3.

## Item analysis

Table 3 shows the item analysis results. Four items (items 1, 6, 11, and 12) met the exclusion criteria for population distribution; one item (item 11) met the exclusion criteria for kurtosis and skewness; four items (items 2, 3, 12, and 13) met the exclusion criteria for inter-item correlation. However, we retained items 2 and 3: the item-total correlation of these items were the second and third highest and were considered important items in the scale. Thus, five items (items 1, 6, 11, 12, and 13) were excluded and 8 items (items 2–5 and 7–10) were subjected to factor analysis.

**Table 3. Item analysis of the "Parental Self-Efficacy Scale for Preventing Challenging Behavior in Children With Autism Spectrum Disorder".** n = 260.

| | Item | Population distribution (%) a | Kurtosis /skewness b | | Item difficulty (%) c | Inter-item correlation d | Item-total correlation e | Good-poor analysis f | Exclusion |
|---|---|---|---|---|---|---|---|---|---|
| 1 | I can watch over my child's growth and development with a caring eye. | **90.7** | -0.47 | 0.18 | 0.40 | — | 0.54 ** | 0.00 | × |
| 2 | I can communicate to my child that I am keeping a caring eye on him/her. | 82.6 | -0.58 | -0.06 | 0.40 | + | 0.60 ** | 0.00 | |
| 3 | I can communicate to my child that I sympathize with him/her. | 83.1 | -0.55 | 0.38 | 0.00 | + | 0.59 ** | 0.00 | |
| 4 | I can ascertain what my child wants to do. | 78.8 | -0.47 | 0.45 | 0.00 | — | 0.66 ** | 0.00 | |
| 5 | I can tell my child about the schedule and plans for the day. | 70.4 | -0.45 | -0.59 | 0.00 | — | 0.51 ** | 0.00 | |
| 6 | I can respond immediately to changes in my child's health. | **85.8** | -0.44 | 0.45 | 0.00 | — | 0.54 ** | 0.00 | × |
| 7 | I can create regular daily routines for my child. | 72.7 | -0.34 | -0.29 | 0.00 | — | 0.50 ** | 0.00 | |
| 8 | I can create places where my child feels comfortable. | 78.0 | -0.20 | 0.18 | 0.40 | — | 0.57 ** | 0.00 | |
| 9 | I can reduce stimulations that my child does not like. | 76.5 | -0.34 | 0.52 | 0.00 | — | 0.52 ** | 0.00 | |
| 10 | I can create opportunities for my child to interact with people in a way that is appropriate for his/her growth and development. | 67.3 | -0.34 | 0.07 | 1.20 | — | 0.58 ** | 0.00 | |
| 11 | I can consult people around me when I need advice related to my child. | **90.0** | **-1.12** | **1.04** | 0.00 | — | 0.43 ** | 0.00 | × |
| 12 | I can tell people close to my child what he/she likes and dislikes. | **89.6** | -0.80 | 0.03 | 0.00 | + | 0.55 ** | 0.00 | × |
| 13 | I can discuss with people close to my child the types of play that facilitate the growth and development of my child. | 83.7 | -0.59 | 0.11 | 0.80 | + | 0.49 ** | 0.00 | × |

** : $p < 0.001$

Exclusion criteria of the item analyses.

[a] : Percentage of 'agree' and 'agree a little' is over 85% of the sample.

[b] : Kurtosis and skewness are over ±1.0 of the sample.

[c] : Percentage of non-respondents is over 5%.

[d] : Correlation between each item is over 0.6.

[e] : Correlation coefficient between the item and the total of all the items (but with exception of the item) is less than 0.3.

[f] : Difference of the average score between most high-scoring group and most low-scoring group is not significant difference($p \geq 0.05$).

**Table 4. Exploratry factor analysis of the PASEC.** n = 130.

| Initial version scale item no. | Item/〈Factor〉 | empowerment of children's sociality | optimization of children's environment |
|---|---|---|---|
| 2 | I can communicate to my child that I am keeping a caring eye on him/her. | **0.88** | -0.13 |
| 3 | I can communicate to my child that I sympathize with him/her. | **0.62** | 0.17 |
| 4 | I can ascertain what my child wants to do. | **0.67** | 0.10 |
| 8 | I can create places where my child feels comfortable. | 0.10 | **0.58** |
| 9 | I can reduce stimulations that my child does not like. | -0.09 | **0.89** |
| 10 | I can create opportunities for my child to interact with people in a way that is appropriate for his/her growth and development. | 0.07 | **0.55** |
| | Cumulative contribution (%) | 41.2 | 53.0 |
| | Factor correlation coefficients (r)  Factor 1 | 1.00 | |
| | Factor 2 | 0.52 | 1.00 |

## Factor analysis

The results of exploratory factor analysis appear in Table 4. The eigenvalues were 3.515 for one factor, 1.192 for two factors, and 0.976 for three factors; the eigenvalues and scree plot suggested a one-factor or two-factor model. We repeated exploratory factor analysis with promax rotation until the factor loadings exceeded 0.5: the difference in factor loadings between each factor became clear, and the factors became theoretically most explicable. As a result, we excluded items 5 and 7 because the factor loading did not exceed 0.5 in any analysis. Excluding items with a loading of less than 0.5 resulted in a two-factor solution; we extracted six items on two factors for a final version of the scale. Factor 1 included three items (items 2–4) interpretable as "empowerment of children's sociality," that is a belief that parents can empower their children's sociality. Factor 2 included three items (items 8–10) interpretable as "optimization of children's environment," that is a belief that parents can optimize their children's environment. The factor loadings were greater than 0.5 for each factor. The cumulative contribution of the two factors explained 53.0% of the variance. Moreover, the correlation coefficient between the two factors was 0.52 (Table 4).

## Internal consistency and validity of the final scale

We entered one factor as a latent factor in a confirmatory factor analysis model. The model fit showed GFI = 0.892, AGFI = 0.749, CFI = 0.885, and RMSEA = 0.172; these results did not represent a good data-model fit. We entered those two factors as latent factors in a confirmatory factor analysis model. The model fit showed GFI = 0.981, AGFI = 0.944, CFI = 0.999, and RMSEA = 0.019; these results satisfied the appropriate criteria in all subjects (Fig 1).

We recognized an error correlation between items 9 and 10 and hypothesized as follows. Items 9 and 10 have in common optimizing the environment for child growth and development. However, item 9 relates to reducing stimuli for children with ASD: "I can reduce stimulations that my child does not like." In contrast, item 10 relates to stimulating children with ASD: "I can create opportunities for my child to interact with people in a way that is appropriate for his/her growth and development." Thus, a negative correlation between the errors reflected the reverse perception of the common behavior of the parents. Thus, construct

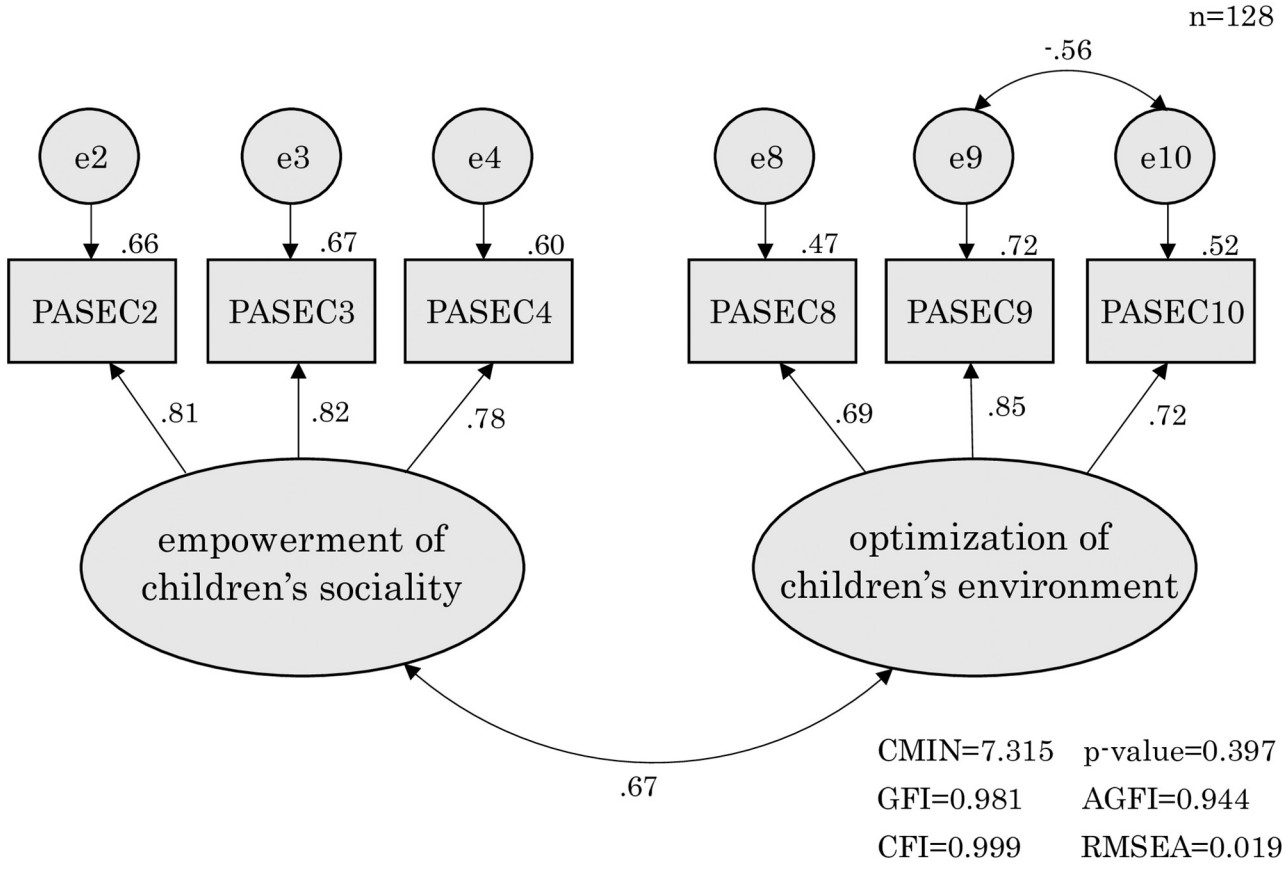

**Fig 1. The confirmatory factor analysis of the PASEC (final version).**

validity was demonstrated. Cronbach's alpha coefficients were 0.81 for factor 1, 0.73 for factor 2, and 0.82 for the whole scale.

Pearson's correlation analysis showed correlations between the total PASEC score and the total score of the self-efficacy subscale of the PSOC, ABC-I, and K6. The PASEC displayed a high positive correlation with the self-efficacy subscale of the PSOC ($r = 0.52$, $P < 0.001$). The PASEC showed a moderate negative correlation with the ABC-I ($r = -0.34$, $P < 0.001$) and with the K6 ($r = -0.30$, $P < 0.001$).

## Discussion

Among the children with ASD, the proportion of boys was 75.4%. This is almost the same figure as that reported in a profile of participants in a study using receipt data from Japan Medical Data Center Co., Ltd. (JMDC): 76.1% were found to be boys [41]. We found the degree of challenging behavior (ABC-I total score) in the children with ASD to be 10.3 ± 8.7. That is similar to the result in one study that measured challenging behavior in children with developmental disabilities, including preschool children with ASD: the parent rating was 12.8 ± 9.8; the teacher rating 10.7 ± 8.0 [42]. The prevalence of depression among the parents participating in the present survey was 45.2%. That is high compared with 32.7% observed in a previous study in Japan [43], and is similar to the 48.6% found in an Oman study [44]. The differences may be accounted for by the fact that the present study and the one conducted in Oman [44] focused on parents of children with ASD; the earlier study in Japan [43] examined parents of children

with various developmental disabilities. One investigation [45] reported that parents of children with ASD suffer more from depression than parents of children with other developmental disabilities. Thus, the sample was representative of the population of parents of children with ASD.

The PASEC demonstrated adequate reliability and validity according to confirmatory factor analysis. The originality of this scale is based on the following two points. The first point is the measurement target. Existing parenting self-efficacy scales primarily target parents of typically developing children [18]; the scale developed in the present study targets parents of children with ASD. The second point is the measurement concept. Wittkowski [18] stated that there are three types of self-efficacy: general, domain-, and task-specific. Hitherto, parenting self-efficacy scales have been domain-specific, such as parenting of parents [19]; the self-efficacy of the scale developed in the present study is task-specific, such as preventing challenging behavior.

The first factor of the PASEC includes items that reflect self-efficacy, allowing parents to empower their children's sociality. Matson et al. [2] pointed out that social interaction deficits of children with ASD are related to challenging behavior; children with ASD may express their own feelings and wills through challenging behavior. Therefore, challenging behavior can be prevented if parents understand the children's feelings and wills in advance and respond accordingly. If parents check their children's feelings and wills on a daily basis, the children may understand how to convey their feelings and wills and actually communicate rather than resorting to challenging behavior [21].

The second factor of the PASEC includes items that reflect a self-efficacy that allows parents to optimize their children's environment. Children with ASD have unique sensibilities, such as hypersensitivity, and feel stimulation that other children do not [14]. Therefore, challenging behavior can be prevented by removing in advance stimuli that such children have difficulty dealing with. In addition, challenging behavior should not impede the opportunity to promote the growth and development of children who display such behavior [3]. Parents would create opportunities for their children to interact with people other than themselves, so that children with communication disabilities and poor ability in dealing with changing situations can develop social skills and become accustomed to different environment.

This study has a few limitations. First, this was a cross-sectional study, and the predictive validity is unclear. Bandura provided a theoretical description of the relationship between self-efficacy and behavior [15, 46]. However, it is necessary to conduct a longitudinal study to determine whether a person with high self-efficacy takes behavior for preventing challenging behavior. Second, the participants in this study were limited to parents who used a developmental support center; the children of those parents were considered to have strong ASD characteristics. However, ASD is a spectrum disorder and contains a gray zone that does not reach diagnostic criteria; thus, future research would need to be conducted in government offices, which are often used by parents of children with ASD in the gray zone. In this study, we defined children with ASD in the gray zone as those who had not been diagnosed but showed the characteristics of ASD. Third, the assessment of ASD was made by the managers of child development support centers and the children's parents. However, it was clear that the centers in this study were used by children with developmental disabilities or particular developmental characteristics. In addition, in real-life situations, support for children with ASD and their parents may be recommended by caregivers or though the self-report of the parents. Thus, it would appear that this study reflected the actual situation, which is a strength.

## Supporting information

**S1 Appendix. PASEC English version.**
(PDF)

**S2 Appendix. PASEC Japanese version.**
(PDF)

## Acknowledgments

We would like to thank all participants in this study. We would also like to express the greatest appreciation to Assistant Professor K Shiratani, E Ito, and all members of the Department of the Community Health Nursing, the Graduate School of Medicine, Yokohama City University, for providing valuable advice throughout the study process.

We thank the Edanz Group (https://en-author-services.edanzgroup.com/ac) for editing a draft of this manuscript.

## Author Contributions

**Conceptualization:** Yuri Kabashima, Etsuko Tadaka, Azusa Arimoto.

**Data curation:** Yuri Kabashima, Azusa Arimoto.

**Formal analysis:** Yuri Kabashima, Azusa Arimoto.

**Funding acquisition:** Etsuko Tadaka.

**Investigation:** Yuri Kabashima, Azusa Arimoto.

**Methodology:** Yuri Kabashima, Azusa Arimoto.

**Project administration:** Etsuko Tadaka.

**Resources:** Etsuko Tadaka.

**Software:** Yuri Kabashima.

**Supervision:** Etsuko Tadaka.

**Validation:** Yuri Kabashima, Etsuko Tadaka, Azusa Arimoto.

**Visualization:** Yuri Kabashima.

**Writing – original draft:** Yuri Kabashima, Azusa Arimoto.

**Writing – review & editing:** Etsuko Tadaka.

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
