## [Decision Letter · Decision Letter 0]

23 Jun 2020

PONE-D-20-10080

Development of the Parental Self-Efficacy Scale for Preventing Challenging Behaviors in Children with Autism Spectrum Disorder

PLOS ONE

Dear Dr. Kabashima,

Thank you for submitting your manuscript to PLOS ONE. After careful consideration, we feel that it has merit but does not fully meet PLOS ONE’s publication criteria as it currently stands. Therefore, we invite you to submit a revised version of the manuscript that addresses the points raised during the review process.

The three reviewers addressed a number of major and minor concerns about your manuscript. Please revise your manuscript carefully. 

We look forward to receiving your revised manuscript.

Kind regards,

Kenji Hashimoto, PhD

Academic Editor

PLOS ONE

Journal Requirements:

Reviewers' comments:

Reviewer's Responses to Questions

**Comments to the Author**

1. Is the manuscript technically sound, and do the data support the conclusions?

Reviewer #1: Partly

Reviewer #2: Partly

Reviewer #3: Yes

2. Has the statistical analysis been performed appropriately and rigorously? 

Reviewer #1: I Don't Know

Reviewer #2: No

Reviewer #3: I Don't Know

3. Have the authors made all data underlying the findings in their manuscript fully available?

Reviewer #1: No

Reviewer #2: Yes

Reviewer #3: Yes

4. Is the manuscript presented in an intelligible fashion and written in standard English?

Reviewer #1: No

Reviewer #2: Yes

Reviewer #3: Yes

5. Review Comments to the Author

Reviewer #1: The authors concluded that PASEC (Parental Self-Efficacy Scale for Preventing Challenging Behaviors in Children) demonstrated adequate reliability and validity to assess parents’ self-efficacy for preventing challenging behavior for children with ASD (Autism Spectrum Disorders).

They also concluded that the scale can help to prevent challenging behavior and contribute to quality of life for children with ASD and their parents.

The strength of this study is total number of 260 parents.

However I have a few questions in this study.

#1: Why did you assess parents as depression when K6 was over 5?

#2: What do you think about abuse problems in this study? (Although you stated the limitation of gray-zone ASD.)

#3: Related to #2, would you tell me your definition of gray-zone ASD?

Reviewer #2: In this study, kabashima et al, focused on self-efficacy of parents with ASD children and developed a new scale.

Major points

Certainly, a self-efficacy scale for parents of ASD children may not exist. However, in order to appeal the need for a scale, it is necessary to assume that the self-efficacy of ASD parents and the self-efficacy of TD parents are significantly different. For example, I think there is a need for the fact that multi-population simultaneous analyzes using currently available self-efficacy measures have been reported to show different structures in the ASD and TD groups. The convergent validity with self-efficacy is shown, but it is not possible to prove the necessity of new scale by itself. In the current situation of overflowing scale, it is considered most necessary to prove that a new scale can measure a new concept that cannot be measured by existing scales when creating a new scale.

Validation is related to self-efficacy, but isn't this enough to show the efficacy of the new scale? If the scale is to contribute to the prevention of challenging behaviors, the association between the scale and the challenging behavior should be included in the validation. Why isn't there an analysis in the paper, even though the study also obtained a scale of challenging behavior? If the relationship with the parent's QOL is also explained, shouldn't it also be shown with the QOL? The concept of measurement is different between K6 and QOL, but I think it would have been better to show only the relationship between mental health and the new scale.

Minor points are as follows.

157 Of those subscales, we used only Irritability (ABC-I) because we judged that the subscale represented a characteristic state of challenging behavior noticed by parents.

Evidence is needed to make this judgment. Please explain with reference materials.

162 The subscale had a Cronbach’s alpha of 0.92.

171 This PSOC scale had a Cronbach’s alpha of 0.79; the self-efficacy subscale had a Cronbach’s alpha of 0.76.

Is this reliability coefficient calculated in this research? Is it a prior research? If it is a previous research, please add a reference.

180 We undertook item and exploratory factor analyses to investigate the reliability and convergent validity of the scale.

I could not understand the meaning of the sentence. Isn't exploratory factor analysis an analysis to examine factor structure, not to verify reliability or convergent validity?

198 We evaluated a correlation of ≥0.50 as adequate.

The evidence thus considered should be added as a reference.

218 The eigenvalues and scree plot suggested a one- or two-factor model.

Is there a transition in The eigen values somewhere in this paper? If not, it should be shown.

233 We entered those two factors as latent factors in a confirmatory factor analysis model. The　model fit showed GFI = 0.981, AGFI = 0.944, CFI = 0.999, and RMSEA = 0.019; it satisfied the appropriate criteria in all subjects.

Two-factor solutions are subject to confirmatory factor analysis, but one-factor solutions should also be subjected to confirmatory factor analysis to examine the degree of improvement in model fit.

In the confirmatory factor analysis, it is assumed that there is a correlation between e9 and e10. Is there any explanation about this? Unless you have a hypothesis, you should not assume correlations between errors in confirmatory factor analysis.

Reviewer #3: The authors developed a novel self-reported questionnaire to assess parental self-efficacy scale to prevent challenging behaviors in children with autism spectrum disorder (ASD) and evaluated its reliability and validity.

The current study seems to be well designed. The manuscript was well written. The novelty and relevance of the study aim is clearly stated. The reviewer has a few minor concerns which should be addressed before the publication.

1. Please clarify the method to confirm the diagnosis of ASD in the children whose parents participated in the current study. If the diagnosis was just confirmed by the parent's self-report, a potential limitation raised from the decision not to utilize any other confirmation of diagnosis by diagnostic tools for ASD should be addressed in the discussion section.

2. In page 8, it was described that the informed consent letters were sent to candidates for participants. However, I cannot find the methods how to obtain the informed consent or how to assess the ability to consent throughout the manuscript. Please clarify these processes regarding informed consent in the current study.

6. PLOS authors have the option to publish the peer review history of their article (what does this mean?). If published, this will include your full peer review and any attached files.

Reviewer #1: No

Reviewer #2: No

Reviewer #3: No

---

## [Author Response · Author response to Decision Letter 0]

6 Aug 2020

Response letter

Dr. Kenji Hashimoto

Academic Editor

PLOS ONE 

7 August 2020

Dear Dr. Kenji Hashimoto,

Thank you very much for your e-mail regarding our manuscript, “Development of the Parental Self-Efficacy Scale for Preventing Challenging Behaviors in Children with Autism Spectrum Disorder” (PONE-D-20-10080). We are delighted to hear that it is potentially acceptable for publication in PLOS ONE. Please find attached a revised version of our manuscript.

Your comments and those of the reviewers were highly insightful and enabled us to greatly improve the quality of our manuscript. Below, we present our point-by-point responses to each of the comments made by the reviewers. 

We look forward to hearing from you regarding our resubmission. We would of course be very happy to respond to any further questions or comments.

Sincerely yours,

Yuri Kabashima, MSN, RN, PHN 

Department of Community Health Nursing, Graduate School of Medicine, Yokohama City University, Yokohama, Kanagawa, Japan

Tel: +81 80 4616 9979　　　E-mail: honu.sun.sora@gmail.com

 Author Response

To the comments of Reviewer #1

1) #1: Why did you assess parents as depression when K6 was over 5?

=> Author’s Response

We very much appreciate the reviewer’s comment. One study found that the cutoff point of the K6 was between 4 and 5 [32]. Thus, 5 points or more can be judged as reflecting depression. The K6 was developed as an index of general psychological distress. The above study [32] demonstrated that the K6 showed a high area under the curve (0.93). This was comparable with that of CES-D (0.95), which is an index of depression; that underlines the effectiveness of the K6 as an index of depression. In addition, having six items, the K6 is easy to evaluate and is widely used worldwide. Thus, it allows comparison with results obtained in various countries.

To clarify the above points, we made revisions to line 172 and mentioned the earlier study.

[32] Sakurai K, Nishi A, Kondo K, Yanagida K, & Kawakami N. Screening performance of K6/K10 and other screening instruments for mood and anxiety disorders in Japan. Psychiatry and Clinical Neurosciences. 2011; 65; 434–441. 

doi: 10.1111/j.1440-1819.2011.02236.x.

Page10, Lines 172, Materials and methods

“We used a Japanese version of the K6 to assess the mental health of the parents [31].”

Page10, Lines 175-176, Materials and methods

“The depression cutoff was 5 or higher; thus, if the total score was higher than 5, it was rated as depression [32].”

2) #2: What do you think about abuse problems in this study? (Although you stated the limitation of gray-zone ASD.)

=> Author’s Response 

Thank you very much for this comment. We considered that children with characteristics of ASD (including those in the gray zone) were more likely to display challenging behavior; that challenging behavior could lead to abuse owing to great mental burden on the parents. One population-based study found that children with ASD had a higher risk of maltreatment than those with typical development; the maltreatment was associated with greater likelihood of aggression, hyperactivity, and tantrums for children with ASD [10]. Parents' perceptions of the behaviors in their children with ASD are associated with increased stressful and helpless mental health of parents who cause abuse [6]. Thus, it is important to prevent challenging behavior in children with ASD toward improving mental health in the parents and children and toward primary prevention of child maltreatment and abuse.

 We have added the following references and revised sections in the Abstract and Introduction: 

[6] Miragoli S, Balzarotti S, Camisasca E, & Blasio P. Parents' perception of child behavior, parenting stress, and child abuse potential: Individual and partner influences. Child Abuse & Neglect. 2018; 84; 146-156. doi:10.1016/j.chiabu.2018.07.034

[10] McDonnell CG, Boan AD, Bradley CC, Seay KD, Charles JM, Carpenter LA. Child maltreatment in autism spectrum disorder and intellectual disability: results from a population-based sample. J Child Psychol Psychiatry. 2019;60(5):576-584. doi:10.1111/jcpp.12993

Page3, Lines 48-50, Abstract

“That scale can help prevent challenging behavior; it can contribute to improving the mental health of parents and children with ASD as well as to primary prevention of child maltreatment and abuse.”

Page4, Lines 61-67, Introduction

“One study showed that parents’ perceptions of difficult behavior in their children are associated with increased risk of parental stress and abuse [6]. Parents of children with ASD are more stressed than those with typically developing children and children with other developmental disabilities [7-9]. Children with ASD have a higher risk of maltreatment than those with typical development [10]. Therefore, toward improving the mental health of parents and children with ASD as well as toward primary prevention of maltreatment and abuse, it is important to prevent the challenging behavior of children with ASD.”

Page5, Lines 92-94, Introduction

“Further, improving parents’ self-efficacy to prevent challenging behavior will improve the mental health of both parents and children as well as prevent maltreatment and abuse.”

3) #3: Related to #2, would you tell me your definition of gray-zone ASD?

=> Author’s Response

We are grateful to the reviewer for this comment. In our study, we defined gray-zone ASD children as those who have not been diagnosed with ASD but display its characteristics. To clarify that point, we added the following sentence to the Discussion: 

Page21, Lines 335-337, Discussion

“In this study, we defined children with ASD in the gray zone as those who had not been diagnosed but showed the characteristics of ASD.”

To the comments of Reviewer #2

Major points

1)Certainly, a self-efficacy scale for parents of ASD children may not exist. However, in order to appeal the need for a scale, it is necessary to assume that the self-efficacy of ASD parents and the self-efficacy of TD parents are significantly different. For example, I think there is a need for the fact that multi-population simultaneous analyzes using currently available self-efficacy measures have been reported to show different structures in the ASD and TD groups. The convergent validity with self-efficacy is shown, but it is not possible to prove the necessity of new scale by itself. In the current situation of overflowing scale, it is considered most necessary to prove that a new scale can measure a new concept that cannot be measured by existing scales when creating a new scale.

=> Author’s Response

We thank the reviewer for those comments. In accordance with them, we added a reference and made the following revision: 

[20] Guimond AB, Wilcox MJ, Lamorey SG. The Early Intervention Parenting Self-Efficacy Scale (EIPSES): Scale Construction and Initial Psychometric Evidence. Journal of Early Intervention. 2008; 30(4); 295-320. doi: 10.1177/1053815108320814

Page6-7, Lines 101-122, Introduction

“Through parental self-efficacy scales to prevent challenging behaviors, practitioners can achieve the following: measure parents’ self-efficacy scores for preventing challenging behaviors; identify parents with low self-efficacy scores; and support such parents in improving their self-efficacy to prevent challenging behaviors. Increased parental self-efficacy enables parents to take action to prevent challenging behaviors: that can prevent children’s challenging behaviors, improve the mental health of parents and children with ASD, and primarily prevent child maltreatment and abuse. One parental self-efficacy (PSE) scale, the Early Intervention Parenting Self-Efficacy Scale [20], consists of two factors: Parent Outcome Expectations; and Parent Competence. Parent Outcome Expectations relates to caregivers perceiving themselves as effective in parenting their children. Parent Competence relates to the belief that the outcome of ASD is a function of environmental influences or constraints, such as family background. However, parents of children with ASD have subject-specific challenges that make it difficult to adapt existing versions of the PSE scale owing to differences in concepts. The subject-specific challenges for parents of children with ASD are that the children have communication difficulties; the children are also hypersensitive or insensitive to stimuli from the surrounding environment, which induce challenging behaviors. Thus, children with ASD need more communication experiences to form patterns of communication and develop social skills; to reduce their children’s discomfort, the parents need to adjust the stimuli that are uncomfortable for their children as well as positive stimuli. To that end, we believe that the Parental Self-Efficacy Scale for Preventing Challenging Behaviors in Children with Autism Spectrum Disorder (PASEC) can measure parents’ self-efficacy scores for preventing challenging behaviors. Therefore, a new scale, PASEC, needs to be developed.”

2)Validation is related to self-efficacy, but isn't this enough to show the efficacy of the new scale? If the scale is to contribute to the prevention of challenging behaviors, the association between the scale and the challenging behavior should be included in the validation. Why isn't there an analysis in the paper, even though the study also obtained a scale of challenging behavior? If the relationship with the parent's QOL is also explained, shouldn't it also be shown with the QOL? The concept of measurement is different between K6 and QOL, but I think it would have been better to show only the relationship between mental health and the new scale.

=> Author’s Response

We very much appreciate the reviewer’s suggestions. In response to Reviewer #1, we decided to focus on the mental health of the parents and children and preventing abuse toward preventing challenging behavior. 

We analyzed the correlation between the total ABC-I score (reflecting the degree of challenging behavior) and total PASEC score. We found that the correlation coefficient was r = –0.34 (P <0.001) and showed a negative correlation. We analyzed the correlation between the total K6 score (reflecting the mental state of the parents) and the total PASEC score. We found that the correlation coefficient was r = –0.30 (P <0.001) and displayed a negative correlation. 

 Accordingly, we made the following revisions:

Page3, Lines 48-50, Abstract

“That scale can help prevent challenging behavior; it can contribute to improving the mental health of parents and children with ASD as well as to primary prevention of child maltreatment and abuse.”

Page4, Lines 61-67, Introduction

“One study showed that parents’ perceptions of difficult behavior in their children are associated with increased risk of parental stress and abuse [6]. Parents of children with ASD are more stressed than those with typically developing children and children with other developmental disabilities [7-9]. Children with ASD have a higher risk of maltreatment than those with typical development [10]. Therefore, toward improving the mental health of parents and children with ASD as well as toward primary prevention of maltreatment and abuse, it is important to prevent the challenging behavior of children with ASD.”

Page5, Lines 92-94, Introduction

“Further, improving parents’ self-efficacy to prevent challenging behavior will improve the mental health of both parents and children as well as prevent maltreatment and abuse.”

Page10, Lines 172, Materials and methods

“We used a Japanese version of the K6 to assess the mental health of the parents [31].”

Page12-13, Lines 227-230, Materials and methods

“In addition, we assumed that the total PASEC score and total ABC-I score (indicating the degree of challenging behavior of children) and total K6 score (indicating the mental health of parents) would have a negative correlation. Therefore, we calculated the correlation coefficients to clarify the relationships.”

Page18, Lines 279-283, Results

“Pearson’s correlation analysis showed correlations between the total PASEC score and the total score of the self-efficacy subscale of the PSOC, ABC-I, and K6. The PASEC displayed a high positive correlation with the self-efficacy subscale of the PSOC (r = 0.52, P <0.001). The PASEC showed a moderate negative correlation with the ABC-I (r = –0.34, P <0.001) and with the K6 (r = –0.30, P <0.001).” 

Minor points 

1)157 Of those subscales, we used only Irritability (ABC-I) because we judged that the subscale represented a characteristic state of challenging behavior noticed by parents.

Evidence is needed to make this judgment. Please explain with reference materials

=> Author’s Response

Thank you very much for your comment. In accordance with it, we have added the following reference and made this revision: 

[35] Bearss K, Johnson C, Smith T, Lecavalier L, Swiezy N, Aman M, McAdam DB, Butter E, Stillitano C, Minshawi N, Sukhodolsky DG, Mruzek DW, Turner K, Neal T, Hallett V, Mulick JA, Green B, Handen B, Deng Y, Dziura J, Scahill L. Effect of parent training vs parent education on behavioral problems in children with autism spectrum disorder. JAMA. 2015; 313(15); 1524-1533. doi:10.1001/jama.2015.3150

Page10, Lines 183-187, Materials and methods 

“The item of Irritability (ABC-I) includes destructive and aggressive behavior toward oneself or others as the challenging behavior [4,5]. Destructive and aggressive behavior is commonly recognized by the parents of children with autism as a challenging behavior [17]. The ABC-I has also been used as an indicator of behavioral problems, which are similar to challenging behavior [35].”

2)162 The subscale had a Cronbach’s alpha of 0.92.

171 This PSOC scale had a Cronbach’s alpha of 0.79; the self-efficacy subscale had a Cronbach’s alpha of 0.76.

Is this reliability coefficient calculated in this research? Is it a prior research? If it is a previous research, please add a reference.

=> Author’s Response

We appreciate the reviewer’s comments. The reference number [34] was added to lines 190-191 because Cronbach’s alpha of the subscale in lines 190-191 appeared in a previous study with reference number [34]. Further, we added reference number [37] to lines 200-201 because Cronbach’s alpha of the scale and subscale in lines 200-201 appeared in a previous study with reference number [37].

Page10-11, Lines 190-191, Materials and methods

 “The subscale had a Cronbach’s alpha of 0.92 [34].”

Page11, Lines 200-201, Materials and methods

 “This PSOC scale had a Cronbach’s alpha of 0.79; the self-efficacy subscale had a Cronbach’s alpha of 0.76 [37].”

3) 180 We undertook item and exploratory factor analyses to investigate the reliability and convergent validity of the scale.

I could not understand the meaning of the sentence. Isn't exploratory factor analysis an analysis to examine factor structure, not to verify reliability or convergent validity?

=> Author’s Response

Thank you very much for pointing this out. In light of your comment, we made the following revision: 

Page11-12, Lines 209-210, Materials and methods

“We undertook item analysis to investigate the reliability of the scale and exploratory factor analysis to investigate the factor structure of the scale.”

4) 198 We evaluated a correlation of ≥0.50 as adequate.

The evidence thus considered should be added as a reference.

=> Author’s Response

We certainly appreciate this suggestion. We made the following revision: 

Page12, Lines 227, Materials and methods

“We evaluated a correlation of ≥0.50 as adequate [40].”

5) 218 The eigenvalues and scree plot suggested a one- or two-factor model.

Is there a transition in The eigen values somewhere in this paper? If not, it should be shown.

=> Author’s Response

Thank you very much for this comment. We made the following revision: 

Page16, Lines 250-252, Results

“The eigenvalues were 3.515 for one factor, 1.192 for two factors, and 0.976 for three factors; the eigenvalues and scree plot suggested a one-factor or two-factor model.”

6) 233 We entered those two factors as latent factors in a confirmatory factor analysis model. The model fit showed GFI = 0.981, AGFI = 0.944, CFI = 0.999, and RMSEA = 0.019; these results satisfied the appropriate criteria in all subjects.

Two-factor solutions are subject to confirmatory factor analysis, but one-factor solutions should also be subjected to confirmatory factor analysis to examine the degree of improvement in model fit.

=> Author’s Response

We very much appreciate the reviewer’s comments. Accordingly, we made the following revision: 

Page17, Lines 266-268, Results

“We entered one factor as a latent factor in a confirmatory factor analysis model. The model fit showed GFI = 0.892, AGFI = 0.749, CFI = 0.885, and RMSEA = 0.172; these results did not represent a good data-model fit.”

7) In the confirmatory factor analysis, it is assumed that there is a correlation between e9 and e10. Is there any explanation about this? Unless you have a hypothesis, you should not assume correlations between errors in confirmatory factor analysis.

=> Author’s Response

We are very grateful for this comment. We did indeed have a hypothesis; to clarify that point, we added the following section to the Results: 

Page17-18, Lines 271-277, Results

“We recognized an error correlation between items 9 and 10 and hypothesized as follows. Items 9 and 10 have in common optimizing the environment for child growth and development. However, item 9 relates to reducing stimuli for children with ASD: “I can reduce stimulations that my child does not like.” In contrast, item 10 relates to stimulating children with ASD: “I can create opportunities for my child to interact with people in a way that is appropriate for his/her growth and development.” Thus, a negative correlation between the errors reflected the reverse perception of the common behavior of the parents.”

To the comments of Reviewer #3

1)Please clarify the method to confirm the diagnosis of ASD in the children whose parents participated in the current study. If the diagnosis was just confirmed by the parent's self-report, a potential limitation raised from the decision not to utilize any other confirmation of diagnosis by diagnostic tools for ASD should be addressed in the discussion section. 

=> Author’s Response

Thank you very much for your comments. In accordance with them, we added the following sections to the Methods and Discussion:

Page9, Lines 160-161, Materials and methods

“In this study, assessment of ASD characteristics was not conducted by a doctor or psychologist but by the center managers or the children’s parents.”

Page20, Line 327, Discussion

“This study has a few of limitations.”

Page21, Lines 337-342, Discussion

“Third, the assessment of ASD was made by the managers of child development support centers and the children’s parents. However, it was clear that the centers in this study were used by children with developmental disabilities or particular developmental characteristics. In addition, in real-life situations, support for children with ASD and their parents may be recommended by caregivers or though the self-report of the parents. Thus, it would appear that this study reflected the actual situation, which is a strength.”

2)In page 8, it was described that the informed consent letters were sent to candidates for participants. However, I cannot find the methods how to obtain the informed consent or how to assess the ability to consent throughout the manuscript. Please clarify these processes regarding informed consent in the current study.

=> Author’s Response

We are grateful for this comment. In response, we added the following section to the Methods: 

Page9, Lines 154-160, Materials and methods

“Informed consent for this study was obtained in two steps: at the centers and with the parents. First, we sent a letter to the centers explaining the purpose of the study. We also asked the managers of the centers to assess the ability to consent and to select suitable parents as potential subjects in line with the study’s inclusion criteria. Second, the center managers gave the parents a verbal explanation and handed them request letters, detailing the purpose of the study and stating that participation was entirely free. The managers also handed out questionnaires.”

---

## [Decision Letter · Decision Letter 1]

21 Aug 2020

Development of the Parental Self-Efficacy Scale for Preventing Challenging Behaviors in Children with Autism Spectrum Disorder

PONE-D-20-10080R1

Dear Dr. Kabashima,

We’re pleased to inform you that your manuscript has been judged scientifically suitable for publication and will be formally accepted for publication once it meets all outstanding technical requirements.

Kind regards,

Kenji Hashimoto, PhD

Section Editor

PLOS ONE

Additional Editor Comments (optional):

Reviewers' comments:

Reviewer's Responses to Questions

**Comments to the Author**

1. If the authors have adequately addressed your comments raised in a previous round of review and you feel that this manuscript is now acceptable for publication, you may indicate that here to bypass the “Comments to the Author” section, enter your conflict of interest statement in the “Confidential to Editor” section, and submit your "Accept" recommendation.

Reviewer #1: All comments have been addressed

Reviewer #3: All comments have been addressed

2. Is the manuscript technically sound, and do the data support the conclusions?

Reviewer #1: Yes

Reviewer #3: Yes

3. Has the statistical analysis been performed appropriately and rigorously? 

Reviewer #1: N/A

Reviewer #3: I Don't Know

4. Have the authors made all data underlying the findings in their manuscript fully available?

Reviewer #1: Yes

Reviewer #3: Yes

5. Is the manuscript presented in an intelligible fashion and written in standard English?

Reviewer #1: Yes

Reviewer #3: Yes

6. Review Comments to the Author

Reviewer #1: 1) #1: Why did you assess parents as depression when K6 was over 5?

=> Author’s Response

We very much appreciate the reviewer’s comment. One study found that the cutoff point of the K6 was between 4 and 5 [32]. Thus, 5 points or more can be judged as reflecting depression. The K6 was developed as an index of general psychological distress. The above study [32] demonstrated that the K6 showed a high area under the curve (0.93). This was comparable with that of CES-D (0.95), which is an index of depression; that underlines the effectiveness of the K6 as an index of depression. In addition, having six items, the K6 is easy to evaluate and is widely used worldwide. Thus, it allows comparison with results obtained in various countries.

To clarify the above points, we made revisions to line 172 and mentioned the earlier study.

[32] Sakurai K, Nishi A, Kondo K, Yanagida K, & Kawakami N. Screening performance of K6/K10 and other screening instruments for mood and anxiety disorders in Japan. Psychiatry and Clinical Neurosciences. 2011; 65; 434–441. doi: 10.1111/j.1440-1819.2011.02236.x.

Page10, Lines 172, Materials and methods

“We used a Japanese version of the K6 to assess the mental health of the parents [31].”

Page10, Lines 175-176, Materials and methods

“The depression cutoff was 5 or higher; thus, if the total score was higher than 5, it was rated as depression [32].”

=> Reviewer’s Response

I understand very well.

2) #2: What do you think about abuse problems in this study? (Although you stated the limitation of gray-zone ASD.)

=> Author’s Response

Thank you very much for this comment. We considered that children with characteristics of ASD (including those in the gray zone) were more likely to display challenging behavior; that challenging behavior could lead to abuse owing to great mental burden on the parents. One population-based study found that children with ASD had a higher risk of maltreatment than those with typical development; the maltreatment was associated with greater likelihood of aggression, hyperactivity, and tantrums for children with ASD [10]. Parents' perceptions of the behaviors in their children with ASD are associated with increased stressful and helpless mental health of parents who cause abuse [6]. Thus, it is important to prevent challenging behavior in children with ASD toward improving mental health in the parents and children and toward primary prevention of child maltreatment and abuse.

We have added the following references and revised sections in the Abstract and Introduction:

[6] Miragoli S, Balzarotti S, Camisasca E, & Blasio P. Parents' perception of child behavior, parenting stress, and child abuse potential: Individual and partner influences. Child Abuse & Neglect. 2018; 84; 146-156. doi:10.1016/j.chiabu.2018.07.034

[10] McDonnell CG, Boan AD, Bradley CC, Seay KD, Charles JM, Carpenter LA. Child maltreatment in autism spectrum disorder and intellectual disability: results from a population-based sample. J Child Psychol Psychiatry. 2019;60(5):576-584. doi:10.1111/jcpp.12993

Page3, Lines 48-50, Abstract

“That scale can help prevent challenging behavior; it can contribute to improving the mental health of parents and children with ASD as well as to primary prevention of child maltreatment and abuse.”

Page4, Lines 61-67, Introduction “One study showed that parents’ perceptions of difficult behavior in their children are associated with increased risk of parental stress and abuse [6]. Parents of children with ASD are more stressed than those with typically developing children and children with other developmental disabilities [7-9]. Children with ASD have a higher risk of maltreatment than those with typical development [10]. Therefore, toward improving the mental health of parents and children with ASD as well as toward primary prevention of maltreatment and abuse, it is important to prevent the challenging behavior of children with ASD.” Page5, Lines 92-94, Introduction “Further, improving parents’ self-efficacy to prevent challenging behavior will improve the mental health of both parents and children as well as prevent maltreatment and abuse.”

=> Reviewer’s Response

I understand very well.

3) #3: Related to #2, would you tell me your definition of gray-zone ASD?

=> Author’s Response

We are grateful to the reviewer for this comment. In our study, we defined gray-zone ASD children as those who have not been diagnosed with ASD but display its characteristics. To clarify that point, we added the following sentence to the Discussion:

Page21, Lines 335-337, Discussion

“In this study, we defined children with ASD in the gray zone as those who had not been diagnosed but showed the characteristics of ASD.”

=> Reviewer’s Response

I understand very well.

Reviewer #3: (No Response)

7. PLOS authors have the option to publish the peer review history of their article (what does this mean?). If published, this will include your full peer review and any attached files.

Reviewer #1: No

Reviewer #3: No

---

## [Editor Report · Acceptance letter]

26 Aug 2020

PONE-D-20-10080R1 

Development of the Parental Self-Efficacy Scale for Preventing Challenging Behaviors in Children with Autism Spectrum Disorder 

Dear Dr. Kabashima:

I'm pleased to inform you that your manuscript has been deemed suitable for publication in PLOS ONE. Congratulations! Your manuscript is now with our production department. 

Kind regards, 

on behalf of

Prof. Kenji Hashimoto 

Section Editor

PLOS ONE